# Short-Term Morphofunctional Changes in Previously Treated Neovascular AMD Eyes Switched to Brolucizumab

**DOI:** 10.3390/jcm11195517

**Published:** 2022-09-21

**Authors:** Pasquale Viggiano, Maria Oliva Grassi, Giacomo Boscia, Mariagrazia Pignataro, Giovanni Petruzzella, Enrico Borrelli, Teresa Molfetta, Giovanni Alessio, Francesco Boscia

**Affiliations:** 1Department of Basic Medical Sciences, Neuroscience and Sense Organs, University of Bari “Aldo Moro”, 70121 Bari, Italy; 2Ophthalmology Unit, A.O.U. City of Health and Science of Turin, Department of Surgical Sciences, University of Turin, 10124 Turin, Italy; 3Ophthalmology Department, San Raffaele University Hospital, 20132 Milan, Italy

**Keywords:** age-related macular degeneration, neovascular age-related macular degeneration, optical coherence tomography, retinal disease, brolucizumab

## Abstract

The purpose of the study is to explore the morphofunctional fluctuations in eyes treated for neovascular AMD (nAMD) when treatment is switched from aflibercept or ranibizumab to brolucizumab. A total of 31 eyes of 31 patients with nAMD with type 1 macular neovascularization (MNV) were included. All patients were imaged using spectral domain optical coherence tomography (SD-OCT). The OCT acquisition was performed at the following visits: (i) “T1 visit” corresponding to the last follow-up examination in which an intravitreal injection of aflibercept or ranibizumab was performed before switching to brolucizumab because of the lack of improvement and (ii) “T2 visit” corresponding to the examination performed 1 month after T1, the latter visit corresponding to the day when a switch to brolucizumab injection was performed, (iii) and 1 month after the latter injection “(T3)”. The main outcome measures were: (1) central macular thickness (CMT), (2) choroidal vascularity index (CVI), (3) subfoveal choroidal thickness (CT), and best-corrected visual acuity (BCVA). Functional outcome showed significant differences at each time. Mean ± SD BCVA was 0.43 ± 0.12 LogMAR at T1 and 0.56  ±  0.16 LogMAR at T2 (*p* = 0.038). A significant improvement in BCVA was displayed at T3 (0.34  ±  0.21 LogMAR) as compared with T2 (*p*  = 0.019). CMT analysis showed fluctuations three times. In detail, T2 displayed a thicker CMT in comparison with T1, although not statistically significant (*p* = 0.12). Contrariwise, T3 showed a thinner CMT in comparison with T2 (*p* = 0.002). Analyzing CVI among the three different times, the luminal choroidal area (LCA) and total choroidal area (TCA) showed significantly different values before and after switching to brolucizumab. T2 showed a significant reduction in both vessel lumen and total area compared with T1 (*p* = 0.032 and *p*  =  0.046, respectively). Moreover, T3 showed a greater value of both LCA and TCA in comparison with T2 (*p* = 0.008 and *p*  =  0.01, respectively). CT did not show significant differences at each time (*p* > 0.05). Our results reported early experiences on morphofunctional fluctuations in patients with nAMD who switched to brolucizumab. The anatomical impact of brolucizumab administration appears to result in choroidal vascular enlargement, accompanied by the resolution of subretinal fluid (SRF) and intraretinal fluid (IRF).

## 1. Introduction

Neovascular age-related macular degeneration (AMD) is the primary cause of acquired irreversible vision loss in people aged more than 55 years in the Western world [1,2]. The treatment of neovascular AMD is aimed at reducing exudation caused by macular neovascularization (MNV) [3,4]. Until recently, numerous therapeutic approaches have been attempted, demonstrating partial effectiveness [5,6,7]. In the last decade, the introduction of intravitreal injections of antivascular endothelial growth factor (VEGF) molecules has allowed for expanding the repertoire of treatments for neovascular AMD, becoming the first-line therapy. The main anti-VEGF drugs include aflibercept (Eylea; Regeneron, Tarrytown, NY, USA, and Bayer HealthCare, Berlin, Germany) [4] and ranibizumab (Lucentis; Genentech, South San Francisco, CA, USA) [8]. Importantly, frequent injections are needed to avoid inadequate efficacy [9,10].

The introduction of brolucizumab (Beovu; Novartis, East Hanover, NJ, USA) increased the number of therapeutic choices [11]. In detail, brolucizumab is an antibody fragment that inhibits the VEGF-A isoform. This molecule is characterized by a smaller structure (26 KDa), allowing an increased concentration of the drug within the retina and improved stability [12]. Trial data (HAWK and HARRIER) have demonstrated that brolucizumab every 3 months is noninferior to fixed-dose aflibercept with respect to the change in best-corrected visual acuity (BCVA) from baseline to week 48. Moreover, the brolucizumab treatment seems to be more efficacious in maintaining the resolution of MNV activity markers, including intraretinal fluid (IRF) or subretinal fluid (SRF) [11,12].

Structural optical coherence tomography (OCT) has provided the capability to provide metrics reflecting the retinal and choroidal anatomy (e.g., central retinal thickness [13], choroidal vascularity index [14], and subfoveal choroidal thickness [15]). Accordingly, structural OCT has been employed to observe and quantify the effect of intravitreal anti-VEGF therapy in neovascular AMD. In particular, previous reports described choroidal and retinal changes occurring after ranibizumab or aflibercept therapies and demonstrated significant differences in terms of morphofunctional modifications between these two molecules [16,17]. The characterization of these morphofunctional biomarkers during anti-VEGF therapy may be important for the optimal management of neovascular AMD and for a better comprehension of the different treatments. Considering this, it would be important to characterize the impact of intravitreal injections of brolucizumab on these morphofunctional parameters, reflecting the choroidal and retinal structure.

In the current study, using structural OCT, we evaluated the morphofunctional changes in eyes treated for neovascular AMD when the treatment was switched from aflibercept or ranibizumab to brolucizumab. This could identify helpful biomarkers for disease management.

## 2. Methods

### 2.1. Study Participants

This retrospective study observed the tenets of the Declaration of Helsinki and was approved by the institutional review board of the Department of Basic Medical Sciences, Neuroscience, and Sense Organs, University of Bari “Aldo Moro”. An informed consent waiver was granted to analyze data that were previously collected.

In this study, subjects 55 years of age and older with type 1 MNV in neovascular AMD [18] in at least one eye were identified from the medical records of the Ophthalmology Department at the Department of Basic Medical Sciences, Neuroscience, and Sense Organs, University of Bari “Aldo Moro”. All the patients received anti-VEGF intravitreal injections of ranibizumab or aflibercept and were switched to brolucizumab at the discretion of the treating physician in the absence of an anatomic response (i.e., persistence of OCT signs of exudation) after previous anti-VEGF therapy.

All patients were imaged with an RTVue XR Avanti spectral domain (SD)-OCT (Optovue, Inc., Fremont, CA, USA). The OCT acquisition was performed at the following visits: (i) “T1 visit” corresponding to the last follow-up examination in which an intravitreal injection of aflibercept or ranibizumab was performed before switching to brolucizumab because of the lack of improvement and (ii) “T2 visit” corresponding to the examination conducted 1 month after T1, when a switch to brolucizumab injection was performed, (iii) and 1 month after the latter injection “(T3)”. Furthermore, all patients received a complete ophthalmologic examination, which involved the measurement of Snellen BCVA, IOP, and dilated ophthalmoscopy.

The exclusion criteria included: (i) history of idiopathic or autoimmune uveitis; (ii) infection or inflammation of both eyes; (iii) history of myocardial infarction or cerebrovascular disease within the last 6 months; (iv) the presence of significant cataract; (v) myopia greater than >3.00 diopters; and (vi) any optic neuropathy, including glaucoma.

Moreover, images with a strength index of less than 40 with significant motion artifact or shadowing effect on the choroid were excluded from the analysis.

### 2.2. Imaging Analysis

Subjects underwent SD-OCT imaging using the enhanced depth imaging (EDI) technique (RTVue XR Avanti, version 2016.1.0.26; Optovue, Inc.), between 08:00 and 12:00 a.m.

The main outcome measures were: (1) central macular thickness (CMT), (2) choroidal vascularity index (CVI), and choroidal thickness (CT).

CMT was established as the concentric circles centered on the fovea, having diameters of 1 mm (innermost ring/fovea). CMT was recorded with the Optovue software (RTVue XR Avanti, version 2016.1.0.26; Optovue, Inc.) in the central 1 mm diameter circle.

CVI was investigated using a previously reported algorithm [19,20]. In brief, the EDI-OCT horizontal B-scan passing through the fovea was exported. Afterwards, a binarization of the latter image was performed after delimiting the choroidal boundaries. The borders of the choroid were manually defined as the zone between the Bruch-RPE junction and the sclerochoroidal junction (the upper and lower boundary) and added to the region of interest (ROI) manager. The total choroidal area (TCA) was computed as the total area of the ROI. The images were binarized using “Niblack’s auto local threshold”, and dark pixels were defined as the luminal choroidal area (LCA), while white pixels were defined as the stromal choroidal area (SCA). CVI was obtained as the ratio between LCA and TCA (Figure 1).

Structural B-scan OCT passing through the fovea was binarized after defining the choroidal boundaries. The borders of the choroid were manually defined as the zone between the Bruch-RPE junction and the sclerochoroidal junction (the upper and lower boundary). Dark pixels were defined as the luminal choroidal area (LCA), while white pixels were defined as the stromal choroidal area (SCA). CVI was obtained as the ratio between LCA and TCA.

Choroidal thickness (CT) was assessed by two independent retinal specialists (PV and MOG) with the OCT software. Interobserver agreement (average) was found to be excellent in the CT assessment (0.89 (confidence interval, 0.86–0.90)). CMT, CVI, and CT analysis was performed at each time point (T1, T2, and T3).

### 2.3. Statistical Analysis

To detect departures from normality distribution, a Shapiro–Wilk test was performed for all variables. Means and standard deviation (SD) were computed for all quantitative variables. An ANOVA test for repeated measures was used to compare quantitative variables between following visits. Statistical calculations were performed using Statistical Package for Social Sciences (version 20.0, SPSS Inc., Chicago, IL, USA). The chosen level of statistical significance was *p* < 0.05.

## 3. Results

### 3.1. Characteristics of Subjects Included in the Analysis

A total of 31 eyes of 31 Caucasian patients with neovascular AMD were included in the study. Eighteen patients were females and 13 patients were males. The mean age was 72.5  ±  7.5 (range, 52–85 years). The switched patients had received an average of 10.2  ± 8.34 (range 3–22 years) of intravitreal injections of anti-VEGF drugs prior to switching to brolucizumab. No cases of intraocular inflammation, such as AC cells or vasculitis, were detected. The characteristics of the subjects included in the analysis are summarized in Table 1.

### 3.2. Functional Outcome

Mean ± SD best-corrected visual acuity (BCVA) was 0.43 ± 0.12 LogMAR at T1 and 0.56  ±  0.16 LogMAR at T2 (*p* = 0.042). A significant improvement in BCVA was displayed at T3 (0.56  ±  0.16 LogMAR and 0.34  ±  0.21 LogMAR, *p* = 0.021). A graphical representation of the BCVA changes is shown in Figure 2.

### 3.3. Anatomical Outcome

#### 3.3.1. Central Macular Thickness (CMT) Analysis

Significant changes in CMT were displayed among the different study visits. In detail, CMT at T2 was thicker in comparison with T1 values, although this difference did not reach statistical significance (290.4 ± 65.6 μm and 269.1 ±  46.8 μm, *p* = 0.16). On the contrary, CMT at T3 was significantly thinner as compared with T2 values (236.8 ± 32.3 μm and 290.4 ± 65.6 μm, *p* = 0.013) (Figure 3).

#### 3.3.2. Choroidal Analysis

At T2, a significant reduction in both vessel lumen and total area was displayed, as compared with T1 (*p* = 0.034 and *p* = 0.046, respectively). Moreover, both LCA and TCA were increased at T3 as compared with T2 values (*p* = 0.011 and *p* = 0.01, respectively). No differences in SCA and CVI were detected among different visits (*p* > 0.05 in all the comparisons). Similarly, the CT did not show significant differences at each time (*p* > 0.05 in all the comparisons) (Table 2).

## 4. Discussion

In this study, using CMT, CVI, and CT analysis, we analyzed the anatomical fluctuations in eyes treated for neovascular AMD when the treatment was switched from aflibercept or ranibizumab to brolucizumab.

The long-term neovascular AMD management associated with the maintenance of visual acuity still represents an unmet need. The recent approval of brolucizumab has offered a viable treatment option [21]. However, limited data are available on its impact on clinical practice [22,23]. Here, we describe functional and anatomical early changes after the switch to brolucizumab in clinical routine. Our results indicate that switching to brolucizumab may represent an option particularly for morphological effects in neovascular AMD previously treated with multiple injections of anti-VEGF without sufficient fluid resolution in various anatomical compartments. A significant reduction in CMT was observed at T3, proving an encouraging response on morphological signs of disease activity. In accordance with our study, Bulirsch et al. [24] showed improvement in anatomical outcomes in 63 eyes of 57 patients with neovascular AMD who switched to brolucizumab. Other reports also revealed beneficial improvement of various OCT characteristics at first visit following the switch to brolucizumab [21,25]. Furthermore, our findings displayed significant BCVA changes at T3 compared with T2, highlighting effective functional improvements after the first brolucizumab injection. Likewise, Matsumoto et al. [26] showed significant BCVA improvement 1 month after the first injection of brolucizumab in 36 eyes with neovascular age-related macular degeneration associated with type 1 choroidal neovascularization.

This study was intended to determine the morphological effect of switching to brolucizumab. For this reason, using a novel OCT-based parameter (CVI), we quantified specific changes in the vascular and total area of the choroid. In detail, T3 (i.e., 1 month after brolucizumab injection) was characterized by a significant LCA and TCA increase, and significant choroidal changes were detected between T2 and T1. Therefore, based on our results, a significant increase in both vascular and total choroidal areas (i.e., LCA and TCA) was disclosed in patients who switched from aflibercept or ranibizumab to brolucizumab. Additionally, assuming that brolucizumab is known to have increased intraocular inflammation as compared with other anti-VEGFs [27], our result of increased LCA after switching to brolucizumab might be the result of a subclinical inflammation in the choroid, even in the absence of clinical signs of choroidal inflammation. Agarwal et al. [28] reported that LCA increases were commonly observed during the active disease secondary to ocular inflammation in patients with tubercular multifocal serpiginoid choroiditis. Furthermore, the vascular choroidal area (LCA) is the major constituent of the TCA (60%–65%), and LCA measurements are not affected as much as TCA by factors such as diurnal variation, IOP, refractive errors, axial length [29,30].

Our findings could be important for monitoring neovascular AMD switching to brolucizumab. Other reports revealed that the loading doses of aflibercept or ranibizumab for treatment-naïve AMD caused a decrease in choroidal thickness [31,32]. An OCT multicenter study [33] measured the subfoveal CT after the loading doses of brolucizumab for treatment-naïve eyes, reporting that the CT decrease was greater than that reported for other anti-VEGF agents. It is noteworthy that it needs to be considered that the authors studied only SCT (subfoveal choroidal thickness) alterations of 73 eyes affected by different subtypes of neovascular AMD. More recently, Pellegrini et al. [34] found that choroidal thickness and vascularity significantly decreased after treatment with aflibercept in an nAMD eye. Using OCT-based parameters (CMT, CVI, and CT), this is the first study to investigate short-term morphological changes after switching to brolucizumab. The anatomical impact of brolucizumab administration may result in choroidal vascular enlargement, accompanied by the resolution of SRF and IRF. Therefore, we might speculate a reasonable efficacy of fluid control achieved with brolucizumab compared with other anti-VEGF agents.

Our study has limitations that should be considered when interpreting our findings. A main limitation is that we did not include a control group. Furthermore, our sample size was relatively small, and the study design was cross-sectional. The follow-up period was short. Another possible limitation of the study is that we identified choroidal boundaries manually. Therefore, measurements are potentially subject to intraobserver variability. Another limitation is that OCT B-scans performed at the following visits may be slightly misaligned in patients with macular disorders [35], even with the employment of the OCT follow-up tracking. Even with this limitation, several studies have investigated the CVI in patients with macular disorders. On the other hand, also the strengths of our study should be kept in mind. For each patient, we examined morphofunctional fluctuations after switching to brolucizumab and established significant short-term visual and anatomical changes. Furthermore, CVI investigation might be influenced by several factors. For this reason, we used a previously reported and validated algorithm [20].

In conclusion, our study reports the early morphofunctional changes occurring after brolucizumab treatment in patients with neovascular AMD previously treated with other anti-VEGF molecules (i.e., aflibercept or ranibizumab). We demonstrated that this treatment has a significant morphofunctional impact on these eyes, our results providing new insights for a better understanding of the role of brolucizumab in the treatment of exudative AMD. Additionally, switching to brolucizumab is a valid option, and other studies with a long-term follow-up are needed to verify the clinical significance of our findings. Further prospective studies are required to confirm the results of the current study and elucidate the long-term effect of brolucizumab on the choroidal circulation.

## Figures and Tables

**Figure 1 jcm-11-05517-f001:**
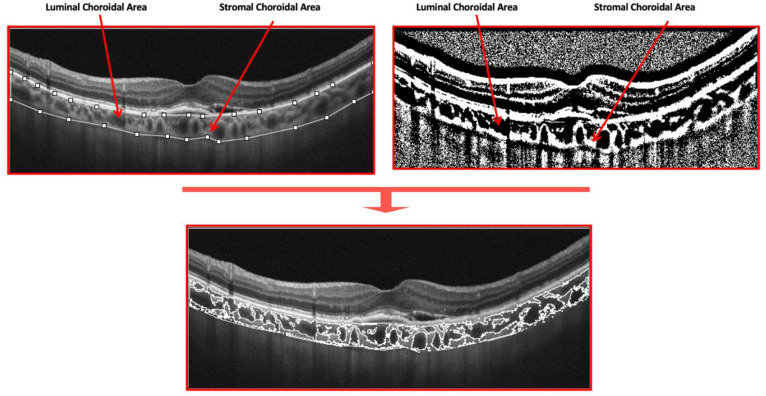
Illustration of the method used to analyze choroidal vasculature from spectral-domain OCT images of an eye with neovascular AMD.

**Figure 2 jcm-11-05517-f002:**
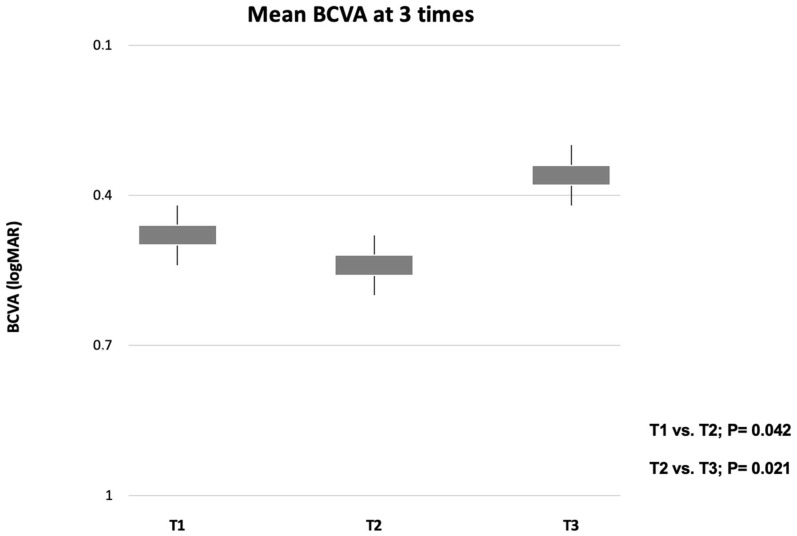
Box and whisker plots showing analyzed BCVA measurements at each time. The ends of the whiskers represent the minimum and maximum values.

**Figure 3 jcm-11-05517-f003:**
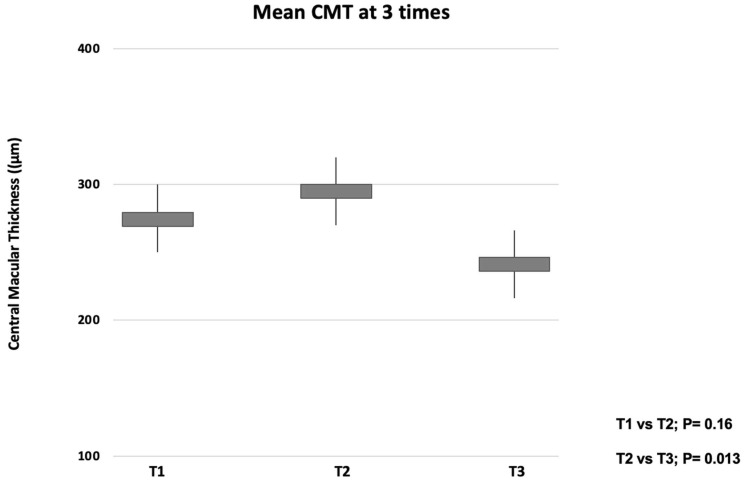
Box and whisker plots showing analyzed central macular thickness (CMT) measurements at each time. The ends of the whiskers represent the minimum and maximum values.

**Table 1 jcm-11-05517-t001:** The clinical characteristics of the subjects included in the analysis. Quantitative data are presented as mean ± SD (standard deviation).

Variables	MNV Type 1 Eyes (*n* = 31)
Age (years)	72.5 ± 7.5
Gender (male,%)	13 (41%)
No. of injections before switch	10.2 ± 8.34
No. of noninfectious intraocular inflammations (IOIs)	N/A

**Table 2 jcm-11-05517-t002:** Structural OCT variables and BCVA changes tested at T1, T2, and T3.

	T1	T2	T3
**CMT (μm)**	269.1 ± 46.8	290.4 ± 65.6	236.8 ± 32.3
		*p* = 0.16 ^a^	*p* = 0.013 ^b^
**LCA (mm^2^)**	0.6761 ± 0.1039	0.6418 ± 0.0809	0.6833 ± 0.0859
		*p* = 0.034 ^a^	*p* = 0.011 ^b^
**SCA (mm^2^)**	0.1344 ± 0.0315	0.1333 ± 0.0389	0.1390 ± 0.0374
		*p* = 0.990 ^a^	*p* = 0.49 ^b^
**TCA (mm^2^)**	0.8096 ± 0.1082	0.7756 ± 0.0991	0.8223 ± 0.0854
		*p* = 0.046 ^a^	*p* = 0.01 ^b^
**CVI (%)**	83.3 ± 4.0	82.7 ± 4.7	83.0 ± 4.4
		*p* = 0.391 ^a^	*p* = 0.690 ^b^
**CT (μm)**	245.1 ± 34.8	251.4 ± 41.6	279.8 ± 54.3
		*p* = 0.344 ^a^	*p* = 0.127 ^b^
**BCVA (LogMAR)**	0.43 ± 0.12	0.56 ± 0.16	0.34 ± 0.21
		*p* = 0.042 ^a^	*p* = 0.021 ^b^

Data are presented as mean ± SD. CMT = central macular thickness; LCA = luminal choroidal area; SCA = stromal choroidal area; TCA = total choroidal area; CVI = choroidal vascularity index; CT = choroidal thickness; BCVA = best-corrected visual acuity. ^a^ comparison T1 versus T2, ^b^ comparison T2 versus T3.

## Data Availability

Results are contained within the article.

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
