# Peer review of "Short-Term Morphofunctional Changes in Previously Treated Neovascular AMD Eyes Switched to Brolucizumab"

_jcm, 2022, doi:10.3390/jcm11195517_

Round 1
Reviewer 1 Report
The authors provide an interesting study on the morphological changes in retina and choroid after switching from aflibercept or ranibizumab to brolucizumab. The manuscript is well written.
However, it would have been better to have included a control arm, with choroidal indices of eyes that responded to aflibercept or ranibizumab. This would have helped to support the hypothesis that brolucizumab causes subclinical inflammation.
There's a similar paper on the short-term effects of aflibercept that can be cited and discussed:
"Pellegrini, M., Bernabei, F., Mercanti, A. et al. Short-term choroidal vascular changes after aflibercept therapy for neovascular age-related macular degeneration. Graefes Arch Clin Exp Ophthalmol 259, 911–918 (2021). https://doi.org/10.1007/s00417-020-04957-5"
Page 9 , line 203 "...while no significant choroidal changes were detected between T2 and T1." While the results mention that TCA and LCA decrease significantly from T1 to T2 (Page 7 , line 174 : "At T2, a significant reduction in both vessel lumen and total area was displayed, as compared to T1 (P=0.032 and P=0.046, respectively). "). Kindly check.
Did the authors notice other signs of inflammation like AC cells, vasculitis? It will be worthwhile to mention in the results.
Author Response
Reviewer 1
The authors provide an interesting study on the morphological changes in retina and choroid after switching from aflibercept or ranibizumab to brolucizumab. The manuscript is well written.
RESPONSE: We thank the Reviewer for these kind words of support for our efforts. We know that a control arm would be helpful to support our hypothesis
However, it would have been better to have included a control arm, with choroidal indices of eyes that responded to aflibercept or ranibizumab. This would have helped to support the hypothesis that brolucizumab causes subclinical inflammation.
RESPONSE: Thanks for your comment. We do totally agree that it would’ve been important to include a control group, although this was not part of our study design as we wanted to assess changes in patients shifting to brolucizumab. However, in the revised version we have specified that “a main limitation is that we did not include a control group. However, our aim was to specifically see changes in eyes shifting to brolucizumab
There's a similar paper on the short-term effects of aflibercept that can be cited and discussed:
"Pellegrini, M., Bernabei, F., Mercanti, A. et al. Short-term choroidal vascular changes after aflibercept therapy for neovascular age-related macular degeneration. Graefes Arch Clin Exp Ophthalmol 259, 911–918 (2021). https://doi.org/10.1007/s00417-020-04957-5"
RESPONSE: We thank the Reviewer for highlighting this important point to clarify. We cited and discussed the paper suggested, as follows:
- More recently, Pellegrini et al.[31] have found choroidal thickness and vascularity significantly decreased after treatment with aflibercept in nAMD eye.
Page 9 , line 203 "...while no significant choroidal changes were detected between T2 and T1." While the results mention that TCA and LCA decrease significantly from T1 to T2 (Page 7 , line 174 : "At T2, a significant reduction in both vessel lumen and total area was displayed, as compared to T1 (P=0.032 and P=0.046, respectively). "). Kindly check.
RESPONSE: We apologize to the Reviewer for the lack of clarity. We agree with the Reviewer that we failed to appropriately explain the results in the “Discussion” section. Therefore, we have made the following changes in the revised “Discussion” section.
- In detail, T3 (i.e. 1 month after brolucizumab injection) was characterized by a substantial LCA and TCA increase, as well as significant choroidal changes were detected between T2 and T1.
Thanks to the reviewer for asking clarification on this point.
Did the authors notice other signs of inflammation like AC cells, vasculitis? It will be worthwhile to mention in the results.
RESPONSE: We thank the Reviewer for highlighting this important point to clarify. We didn’t have cases of intraocular inflammation or other inflammation like AC cells or vasculitis. The latter information was provided in table 1 and discussed in the “Results” section, as follows:
- A total of 31 eyes of 31 Caucasian patients with neovascular AMD were included in the study. Eighteen patients were females and 13 patients were males. The mean age was 72.5 ± 7.5 years (range 52–85 years). The switched patients had received an average of 10.2 ± 8.34 (range 3–22 years) of intravitreal injections of anti-VEGF drugs prior to switching to brolucizumab. No cases of intraocular inflammation like AC cells or vasculitis were detected. The characteristics of the subjects included in the analysis are summarized in Table 1.
Reviewer 2 Report
In this article, the authors analyze BCVA, CMT as well as multiple morphological markers for the choroid to describe the efficacy of brolucizumab treatment in patients with an insufficient treatment response to aflibercept or ranibizumab in nAMD.
The manuscript is well-structured and clearly written. Use of the English language could, however, use some editing.
The following concerns should be addressed:
1) description of the study population
1.1) The authors describe that they identified 55 patients from a medical record search. However, only 31 patients were included in the study. These differences should be explained
1.2.) Patients were previously treated with ranibizumab or Aflibercept. Table 1 should contain numbers of patients that received ranibizumab or aflibercept.
2) Statistical Analysis:
2.1.) Authors describe that they tested for normal distribution. However, they do not report the outcome. CVI for examples is presented in %. % data are usually not normally distributed. Hence, a t-test seems not suitable.
2.2.) All values were quantified at three time points. ANOVA analysis might be more appropriate then paired t-tests to account for multiple testing.
3) Data on CMT should be presented in a stand alone graph similar to Figure 2. Box and whisker plot might be helpful to better judge the variance between patients.
4) It is surprising that TCA and CT increase in response to brolucizumab particularly since these results stand in contradiction to previously published data. The discussion should address this conflict more thoroughly and provide potential explanations.
Author Response
Reviewer 2
In this article, the authors analyze BCVA, CMT as well as multiple morphological markers for the choroid to describe the efficacy of brolucizumab treatment in patients with an insufficient treatment response to aflibercept or ranibizumab in nAMD. The manuscript is well-structured and clearly written. Use of the English language could, however, use some editing.
RESPONSE: We thank the Reviewer for these kind words of support for our efforts.
The following concerns should be addressed:
1) description of the study population
1.1) The authors describe that they identified 55 patients from a medical record search. However, only 31 patients were included in the study. These differences should be explained
RESPONSE: We thank the Reviewer for highlighting this important point to clarify. As described in the “Methods” section:
“ In this study, subjects 55 years of age and older with type 1 MNV in neovascular AMD[18] in at least one eye were identified from the medical records of the ophthalmology department at the Department of Basic Medical Sciences, Neuroscience and Sense Organs, University of Bari "Aldo Moro". All the patients received anti-VEGF intravitreal injections of ranibizumab, or aflibercept and were switched to brolucizumab at the discretion of the treating physician in absence of an anatomic response (i.e. absence of OCT sings of exudation) after previous anti-VEGF therapy.”
1.2.) Patients were previously treated with ranibizumab or Aflibercept. Table 1 should contain numbers of patients that received ranibizumab or aflibercept.
RESPONSE: We thank the Reviewer for highlighting this important point to clarify. We have added the number of patients that received ranibizumab or aflibercept. The latter information was provided in table 1.
2) Statistical Analysis:
2.1.) Authors describe that they tested for normal distribution. However, they do not report the outcome. CVI for examples is presented in %. % data are usually not normally distributed. Hence, a t-test seems not suitable.
RESPONSE: Thanks for this comment. As stated in the manuscript, our data were assessed for normal distribution. Tests confirmed the presence of a normal distribution of choroidal variables in our study cohort.
2.2.) All values were quantified at three time points. ANOVA analysis might be more appropriate then paired t-tests to account for multiple testing.
RESPONSE: Thanks for this comment. However, assuming that we included a single group that was assessed at different time points, we felt the better analysis was a paired-samples T-test.
3) Data on CMT should be presented in a stand-alone graph similar to Figure 2. Box and whisker plot might be helpful to better judge the variance between patients.
RESPONSE: We apologize with the Reviewer for the lack of clarity. We agree with the Reviewer. For this reason, we have presented CMT data using box and whisker plot graph, as showed in the Figure 3.
4) It is surprising that TCA and CT increase in response to brolucizumab particularly since these results stand in contradiction to previously published data. The discussion should address this conflict more thoroughly and provide potential explanations.
RESPONSE: We thank the Reviewer for highlighting this important point to clarify. In agreement with these reports, we found a significant CMT reduction after switch to brolucizumab. On contrary, no significant differences in CT were demonstrated in our study cohort. We would assume that differences in study cohort among studies, as well as our small sample size and short follow-up may have impacted on this result. The publication of our data may justify the following studies clarifying this point. This has been discussed in our paper, as follows
- “Other reports revealed that the loading doses of aflibercept or ranibizumab for treatment-naïve AMD caused a decrease in choroidal thickness.[28,29] An OCT multicenter study,[30] measured the subfoveal CT after the loading doses of brolucizumab for treatment-naïve eyes, reporting that the CT decrease was greater than that reported for other anti-VEGF agents. Noteworthy, it needs to be considered that the authors studied only SCT alterations of 73 eyes affected by different subtypes of neovascular AMD. More recently, Pellegrini et al.[31] have found choroidal thickness and vascularity significantly decreased after treatment with aflibercept in nAMD eye. Using OCT-based parameters (CMT, CVI and CT), this is the first study to investigate short-term morphological changes after switching to brolucizumab. The anatomical impact of brolucizumab administration may result in the more effective resolution of SRF and IRF, in association with choroidal vascular enlargement. Therefore, these results indicate greater efficacy of fluid control achieved with brolucizumab compared with other anti-VEGF agents”.
Reviewer 3 Report
The paper entitled “Short-term morpho-functional changes in previously treated neovascular AMD eyes switched to brolucizumab” is a study based on morpho-functional clinical outcomes in eyes treated for nAMD when switched to brolucizumab. The manuscript is interesting and of potential clinical interest.
It is widely known how neovascular age-related macular degeneration (AMD) causes important irreversible vision loss. Modern treatment options entail intravitreal injections of anti-vascular endothelial growth factor (VEGF). Numerous molecules have been proposed in the past years, thus studies based on the effects of new drugs are of great clinical use when managing these patients, especially in deciding the best intravitreal therapy to use. Functional and structural measures based on BCVA and OCT scans help provide objective results.
The study has been correctly planned and provides objective results. Figures are pertinent and provide good examples of structural results. References are pertinent and based on current literature.
There are, however, a few minor considerations:
- Acronyms need to be defined in the Abstract.
- The main outcome measurements in the study included BCVA, which needs to be added in the Methods section of the Abstract.
- The authors are correct in stating the limitations regarding the small number of patients, short follow-up, and retrospective nature of the study. The lack of a control group reporting a similar cohort using a different agent could be considered.
- Mention regarding future prospective study designs should be added in the Conclusion section.
- The paper can be improved if corrected by a native English-speaking doctor to enhance the flow.
Author Response
Reviewer 3
The paper entitled “Short-term morpho-functional changes in previously treated neovascular AMD eyes switched to brolucizumab” is a study based on morpho-functional clinical outcomes in eyes treated for nAMD when switched to brolucizumab. The manuscript is interesting and of potential clinical interest.
It is widely known how neovascular age-related macular degeneration (AMD) causes important irreversible vision loss. Modern treatment options entail intravitreal injections of anti-vascular endothelial growth factor (VEGF). Numerous molecules have been proposed in the past years, thus studies based on the effects of new drugs are of great clinical use when managing these patients, especially in deciding the best intravitreal therapy to use. Functional and structural measures based on BCVA and OCT scans help provide objective results.
The study has been correctly planned and provides objective results. Figures are pertinent and provide good examples of structural results. References are pertinent and based on current literature.
RESPONSE: We thank the Reviewer for these kind words of support for our efforts.
Acronyms need to be defined in the Abstract.
RESPONSE: We apologize with the Reviewer for the lack of clarity. We have defined the acronyms in the “Abstract” section.
The main outcome measurements in the study included BCVA, which needs to be added in the Methods section of the Abstract.
RESPONSE: We apologize with the Reviewer for the lack of clarity. We have included the BCVA in the Methods section of the Abstract.
The authors are correct in stating the limitations regarding the small number of patients, short follow-up, and retrospective nature of the study. The lack of a control group reporting a similar cohort using a different agent could be considered
RESPONSE: Thanks for your comment. We do totally agree that it would’ve been important to include a control group, although this was not part of our study design as we wanted to assess changes in patients shifting to brolucizumab. However, in the revised version we have specified that “a main limitation is that we did not include a control group. However our aim was to specifically see changes in eyes shifting to brolucizumab”.
Mention regarding future prospective study designs should be added in the Conclusion section.
RESPONSE: We thank the Reviewer for highlighting this important point to clarify. As added in the “Conclusion” section:
“Further prospective studies are required to confirm the results of the current study and elucidate the long-term effect of brolucizumab on the choroidal circulation.”
The paper can be improved if corrected by a native English-speaking doctor to enhance the flow.
RESPONSE: Thanks for your comment. In agreement with the authors, we have corrected and improved the paper to enhance the flow.
Round 2
Reviewer 1 Report
The queries have been well addressed. There are no further comments.
Author Response
We thank the Reviewer for these kind words of support for our efforts.